# Extracellular Vesicles and Renal Fibrosis: An Odyssey toward a New Therapeutic Approach

**DOI:** 10.3390/ijms22083887

**Published:** 2021-04-09

**Authors:** Maja Kosanović, Alicia Llorente, Sofija Glamočlija, José M. Valdivielso, Milica Bozic

**Affiliations:** 1Institute for the Application of Nuclear Energy, INEP, University of Belgrade, 11080 Belgrade, Serbia; maja@inep.co.rs (M.K.); sofija.glamoclija@inep.co.rs (S.G.); 2Department of Molecular Cell Biology, Institute for Cancer Research, Oslo University Hospital, 0379 Oslo, Norway; alillo@rr-research.no; 3Department for Mechanical, Electronics and Chemical Engineering, Oslo Metropolitan University, 0167 Oslo, Norway; 4Vascular and Renal Translational Research Group, Institute for Biomedical Research in Lleida (IRBLleida) and RedInRen RETIC, 25196 Lleida, Spain; josemanuel.valdivielso@udl.cat

**Keywords:** extracellular vesicles, exosomes, cellular communication, renal fibrosis, CKD, therapeutic agents, regeneration, renoprotection, mesenchymal stem cells, cell-free therapeutic

## Abstract

Renal fibrosis is a complex disorder characterized by the destruction of kidney parenchyma. There is currently no cure for this devastating condition. Extracellular vesicles (EVs) are membranous vesicles released from cells in both physiological and diseased states. Given their fundamental role in transferring biomolecules to recipient cells and their ability to cross biological barriers, EVs have been widely investigated as potential cell-free therapeutic agents. In this review, we provide an overview of EVs, focusing on their functional role in renal fibrosis and signaling messengers responsible for EV-mediated crosstalk between various renal compartments. We explore recent findings regarding the renoprotective effect of EVs and their use as therapeutic agents in renal fibrosis. We also highlight advantages and future perspectives of the therapeutic applications of EVs in renal diseases.

## 1. Introduction

Renal fibrosis is the final manifestation of chronic kidney disease (CKD), characterized by progressive destruction of kidney parenchyma and the subsequent loss of renal function. Due to the complexity of this condition and the inability to establish a proper hierarchy of involved mechanisms during the development of renal fibrosis, there is currently no effective antifibrotic therapy in clinical use [1]. Over the past decade, extensive research has focused on the prevention and reversal of kidney fibrosis, with specific emphasis on the regeneration of the injured parenchyma using, among others, different cell-based approaches with multipotent progenitor cells for the reparation of an injured organ [1]. Nevertheless, among diverse approaches proposed as potential therapies for renal fibrosis, the use of extracellular vesicles (EVs) as a cell-free therapeutic approach has now started to be extensively investigated.

In recent years, numerous studies have pointed to EVs as important players in various physiological and pathophysiological processes, including kidney diseases. Apart from being active players in various biological processes [2,3], EVs have been identified as important means of communication alongside the nephron [4]. Furthermore, owing to their versatile characteristics, EVs represent a potential basis for the development of novel therapies [5,6]. Thus, EVs derived from mesenchymal stem cells (MSCs) have emerged as a powerful cell-free therapy for a variety of disease states, renal fibrosis being one of them [7]. 

The current review explores contemporary findings on the functional role of EVs in the pathogenesis of renal fibrosis and signaling messengers involved in EV-mediated crosstalk between various renal and extrarenal cell types. We summarize the current knowledge on the renoprotective role of EVs from diverse sources and their use as therapeutic agents in renal fibrosis, highlighting benefits and new perspectives of therapeutic applications of EVs in renal diseases.

## 2. Extracellular Vesicles: Classification, Biogenesis, and Function

EVs are membrane-limited vesicles that cells release in order to transfer biomolecules to other cells and thus communicate with them [2]. In the last decade, knowledge about EVs vastly expanded, revealing EVs as a novel paradigm in intercellular communication. Numerous studies have demonstrated the involvement of EVs in various physiological and pathophysiological processes, hence highlighting their importance for both functioning of the organism and the design of novel diagnostic tools and therapies [2,3]. 

EVs encompass diverse populations of membrane-limited vesicles released by virtually all cells in the organism. The diversity of EVs refers to their biogenesis, composition, size, and role [8]. 

### 2.1. Classification and Biogenesis of Extracellular Vesicles

EVs are currently classified into two main types based on their biogenesis: exosomes and microvesicles (MVs) (Figure 1). In a wider sense, apoptotic bodies may also be classified as EVs, although their involvement in intercellular communication is far less studied and will not be considered here [2,8]. The process of biogenesis encompasses the selection of cargo molecules, inducing curvature of the membrane and detachment of the vesicle. Although details of these processes are still elusive, significant advances in understanding their basics have recently been made. Hence, the formation of an exosome starts in a multivesicular body (MVB). Two main mechanisms govern this process: endosomal sorting complex required for transport (ESCRT)-dependent mechanism or ESCRT-independent mechanism [8,9]. In the former, ubiquitinated proteins are selected by the members of the ESCRT complex [9,10,11], while in the latter, sphingosine-1-phosphate, Hsc70, and tetraspanins participate in protein selection [9,12]. In addition, the sorting of some proteins may depend on their interaction with lipid raft components [13]. The sorting of miRNAs is based on their interaction with RNA-binding proteins that are targeted to EVs, their abundance in cells, and their sequence [8,14]. Invagination of the MVB limiting membrane is accomplished by the action of phosphatidic acid and sphingolipids [15,16]. This process results in the formation of vesicles in the lumen of MVB—intraluminal vesicles (ILVs) (Figure 1). In order for cells to release ILVs as exosomes, MVB must be targeted to and fused with the plasma membrane. The process of guidance of MVBs to the plasma membrane is governed by RAB proteins [8,17,18], while the fusion of MVB with the plasma membrane is mediated by RAB and SNARE proteins (N-ethylmaleimide-sensitive fusion attachment protein (SNAP) receptors) [19]. 

In contrast to exosomes, microvesicles are formed by direct outward budding of the plasma membrane. Reshaping of the membrane is accomplished by changing its lipid composition at the budding site and disassembly of the cytoskeleton by Ca^++^ ions-dependent enzymes, although it has been shown that specific members of the ESCRT complex are also involved in this process [8,20,21].

Even though modes of biogenesis define two main types of EVs, their other properties are not so distinctive. As for composition, EVs can carry all types of biomolecules (proteins, RNA, lipids, metabolites) involved in a number of processes such as adhesion, metabolism, signal transduction, membrane fusion, organization and trafficking, etc., in addition to molecules involved in EVs biogenesis [2,8]. The composition of a particular EV population also depends on the type and physiological condition of its cell of origin [22]. Different molecules specific for cell type or physiological states, such as specific surface receptors, enzymes, or cell markers, may be packed into EVs making them useful as liquid biopsies [23]. 

All these biogenesis and sorting mechanisms give rise to subpopulations of EVs with partially overlapping cargo [24]. Due to this overlapping and despite the presence of some biogenesis machinery components, no true marker of EV types exists. Although tetraspanins (CD63, CD9, and CD81) are being used as exosome markers, they are present on both exosomes and microvesicles and thus are unable to distinguish these two types of EVs [25]. 

EVs also differ in size and shape. Although it is often stated that exosomes are generally smaller than microvesicles (30–150 nm vs. 100–1000 nm), their sizes in fact overlap and thus cannot be used as a distinctive criterion for the determination of the EV type [26]. As for morphology, most of the EVs of both types have a round shape under an electron microscope, but other forms (elongated and multilayered vesicles) are also observed, adding to EVs heterogeneity. It should be noted that most of the available data have been derived from heterogeneous populations of EVs, comprising both exosomes and microvesicles and their subpopulations since there is no method available for their complete and precise separation [23,27,28]. Additionally, the terms “exosomes” and “microvesicles” have been used in the literature in an inconsistent manner. Nevertheless, in this review, we will use original terms for EVs described by authors in their publications. 

### 2.2. Main Proposed Functions of Extracellular Vesicles

EVs have numerous roles in the body. They are important players in virtually all physiological processes such as placentation, embryonic development, immunity, coagulation, liver homeostasis, nervous system function, tissue repair, and kidney function [2,29]. EVs also participate as key communicators in pathophysiological processes such as cancer, diabetes, atherosclerosis, and endothelial dysfunction, cardiovascular and immune diseases, inflammation, obesity, fibrosis, etc. [3,30,31]. To exert their roles, EVs may use biofluids to reach the target cell and are able to cross the blood–tissue barrier. EVs deliver bio-information to target cells either by the interaction of surface molecules on EVs and the cell or through uptake/fusion of EVs by a target cell [2,32,33], where proteins and RNA can both be effector molecules. 

Aside from their fundamental roles in the organism, EVs can have valuable functions as either diagnostic or therapeutic agents, or the combination of both, known as theranostics tools. EVs reflect the physiological state of the cell they originate from, and they are readily available in biofluids, and as such, they can be used in novel diagnostic platforms [5,34]. This is especially important for the diagnosis of diseases in organs in which tissue biopsy should be avoided whenever possible, such as brain or kidneys. Of note, EVs in urine (uEVs) can originate from different parts of the urinary tract, including the kidney. Carrying diverse molecular cargo from parental cells, uEVs represent an excellent readout of the physiological and pathophysiological state of different parts of the nephron. Thus, uEVs have drawn significant attention as potential biomarkers for diagnostic and prognostic purposes in various renal diseases [35]. For instance, miR-146a from urinary exosomes (uEx) was proposed as a potential biomarker of albuminuria in essential hypertension [36], while miR-29c was suggested as a novel, noninvasive marker for renal fibrosis [37]. miR-145 from uEx was demonstrated to be a promising candidate biomarker for type 1 diabetic nephropathy (DN) [38], while miR-192 was shown to be useful as a predictor of early stage DN [39]. Moreover, CD2AP mRNA in urinary exosomes has been suggested as a noninvasive tool for the detection of changes in kidney function and tubulointerstitial fibrosis, while CCL2 mRNA has been shown as a good predictor of renal function deterioration in IgAN [40]. In addition to miRNAs and mRNAs, various proteins in uEVs were shown to be biomarkers for different renal diseases, among them fetuin-A, ATF3, WT-1, aquaporin-1, etc. [30,41,42,43]. Since the use of EVs as potential diagnostic and prognostic tools for renal diseases has been covered elsewhere [30,41,42,43,44], in this review, we will focus our attention on the potential therapeutic utility of EVs in renal diseases.

Specific properties of EVs such as biocompatibility, biological barrier crossing, selective targeting, and the possibility to be modified make these vesicles a very promising basis for the development of novel therapies [5,6]. Importantly, EVs convey effects of cell therapy without presenting the same hazard as live cells, such as the case for EVs from mesenchymal stem cells (MSC-EVs) [45]. EVs have already been shown to be a beneficial therapy as antitumor vaccines [46], as immune-suppressive agents in graft-vs-host disease [47], and as drug-delivery agents [48,49]. As fundamental tools in cellular communication, EVs hold great promise for both understanding processes that take place in the organism and the design of novel diagnostic and therapeutic tools. 

## 3. Renal Fibrosis

Renal fibrosis is the final manifestation of chronic kidney disease (CKD). It is a complex and dynamic process characterized by an uncontrolled accumulation of extracellular matrix (ECM), leading to a loss of functional renal parenchyma and consequently end-stage renal disease (ESRD), a severe disorder that warrants kidney replacement therapy [50]. The pathogenesis of renal fibrosis involves the participation of diverse molecular pathways and various renal and extrarenal cell types [51]. Thus, changes in different intracellular signaling pathways [52,53], membrane-associated channels [54,55], cytoskeleton and structural proteins [50,56], inflammatory markers [57], and even epigenetic changes [58] have been demonstrated to affect renal fibrosis. It is generally accepted that renal fibrosis initiates as a beneficial reaction to an injury of the renal tissue [59], in which various cell types actively participate in the process of renal healing. However, if the injury is persistent, which is the situation in most progressive kidney diseases, the renal parenchyma fails to regenerate and undergo maladaptive repair, promoting renal fibrosis [60]. 

After an initial insult such as high ambient glucose, protein overload, hypoxia, persistent infection, autoimmune reaction, or chemical insults, injured tubular epithelial cells (TECs) of the kidney display dramatic cellular and molecular rearrangements including cell cycle arrest [61,62], partial epithelial–mesenchymal transition (EMT) [60,63,64], lipotoxicity [65], and autophagy deregulation [51,66,67,68]. Alterations in the epithelial secretome profile of TECs subsequently promote activation and/or proliferation of myofibroblasts [60,69]. Activated myofibroblasts accumulate producing significant amounts of ECM in the renal interstitium and glomeruli [1,51,60,62]. Various chemokines and chemoattractants secreted by TECs trigger the recruitment of immune cells such as macrophages, monocytes, dendritic, and mast cells to the injury sites [70,71]. Arriving at their final destination, these immune cells become further activated, leading to the secretion of inflammatory and fibrogenic cytokines, which additionally contributes to the perpetuation of the remorseless cycle of tissue damage [59,60]. The interplay among epithelial tubular cells, myofibroblasts, endothelial and immune cells is considered to be the major driving force of renal fibrosis [51]. As the process progresses, ECM proteins are deposited in the extracellular compartments, becoming cross-linked and resistant to degradation. The continuation of the process leads to the formation of fibrous scars and distorts the subtle architecture of the kidney, leading to the destruction of renal parenchyma and the loss of kidney function [59].

Regardless of the fact that a range of conditions such as glomerulonephritis, diabetes mellitus, high blood pressure, atherosclerosis, obstructive nephropathy, interstitial nephritis, and polycystic kidney disease can be the major causes of CKD, renal fibrosis is always a key pathway in the progression of nearly every type of chronic renal disease [59,72]. In spite of many efforts, currently, there is no effective treatment for renal fibrosis. The development of new strategies to prevent progression or even reverse the process of renal fibrosis is of essential need in order to halt the progression of CKD. 

## 4. Functional Role of Extracellular Vesicles in Renal Fibrosis

Numerous studies have supported the role of EVs in diverse renal diseases, yet the knowledge of their function in renal fibrosis remains limited. Proteomic analysis of urinary vesicles confirmed that EVs could originate from all segments of the nephron, including glomerular podocytes, proximal tubules, thick ascending limb of Henle, the distal convoluted tubule, and the collecting duct [73]. Apart from being involved in various biological functions such as programmed cell death, angiogenesis, coagulation, inflammation, immunosuppression, growth, and regeneration [74,75], EVs have an important communication function along the nephron [4]. They can mediate the crosstalk between various cell types within the kidney and therefore actively participate in diverse physiological and pathophysiological processes through intra-nephron communication. It has been proposed that signaling carried by EVs might be essential for tubulointerstitial communication during renal fibrosis progression [30] since the injured renal cells may pass EVs containing different molecules to other cells, thus participating actively in renal inflammation and fibrosis (Figure 2). 

Here, we describe the contribution of EVs to different aspects of renal fibrosis. We depict how released EVs regulate/influence the destiny of renal and extrarenal cell types and subsequently the outcome of kidney injury. In addition, wherever possible, we describe signaling messengers from EVs responsible for EV-mediated crosstalk between various renal cellular compartments in renal fibrotic disease.

### 4.1. EVs Mediate Injury of Autochthonous Kidney Cells 

Kidney autochthonous cells principally encompass proximal tubular epithelial cells (PTECs), distal tubular epithelial cells (DTECs), podocytes, and mesangial cells. An expanding body of data points to the role of EVs in the transfer of information between resident kidney cells, which subsequently induces damage of recipient cells and the development of renal fibrosis. It has been demonstrated that injured renal tubular epithelial cells could promote neighboring renal TECs to undergo epithelial–mesenchymal transition by transferring microvesicles (MVs) [76,77]. Namely, TECs-derived MVs carrying miR-21 after being taken by recipient TECs, triggered the activation of the PI3K-Akt pathway, and promoted EMT and progression of renal interstitial fibrosis [76,77]. Another piece of evidence on the role of PTEC-derived vesicles in promoting EMT of kidney epithelial cells is the paper of Qu et al. [78], in which authors showed that MVs secreted from injured PTECs delivered miR-216a to adjacent epithelial cells, inducing changes in renal epithelial phenotype and aggravating renal fibrosis through the PTEN/Akt pathway. A recent study from Jia et al. reported that EVs released from human proximal tubular (HK2) cells treated with high glucose (HG) promoted renal fibrosis by transferring miR-192 to healthy recipient TECs [79]. Using TargetScan and gain- and loss-of-function experiments, the authors confirmed that miR-192 promoted renal fibrosis by targeting glucagon-like peptide 1 receptor (GLP1R). One of the inevitable consequences of kidney aging is the reduction of functional glomeruli and interstitial fibrosis, which is an unstoppable process mainly due to the fact that senescent cells have negative effects on the tissue milieu. Liu et al. 2020 [80] showed that EVs from aging PTECs might play a significant role in promoting senescence and EMT in vicinal PTECs. Namely, human PTECs that underwent senescence and EMT secreted miR-21-containing EVs that induced EMT in recipient cells through targeting the PPARα-HIF-1α signaling pathway and promoting aging-related renal fibrosis. 

It has been demonstrated that crosstalk between podocytes and renal tubular cells has an important role in the development of renal fibrosis. Indeed, exposure of tubular epithelial cells to microparticles (MPs) isolated from cultured media of untreated differentiated human podocytes led to a significant pro-fibrotic response [81]. Podocyte MPs increased p38 and Smad3 phosphorylation and expression of the ECM proteins fibronectin and collagen type IV in cultured PTECs. Antagonism of cell surface scavenger receptor CD36 abrogated all podocyte MP-induced responses in PTECs [81]. Wu et al. 2017 [82] pointed to intercellular crosstalk between cells of the glomerulus as an essential step in the pathogenesis of renal fibrosis. Namely, the authors demonstrated that glomerular endothelial cells (GECs) treated with high glucose (HG) underwent endothelial–mesenchymal transition (EndoMT) and secreted exosomes enriched with TGF-β1 mRNA. After being internalized by podocytes, GEC-derived exosomes mediated EMT and barrier dysfunction through the Wnt/β-catenin signaling pathway. The authors proposed a role for paracrine communication via exosomes as important for the progression of renal fibrosis in diabetic nephropathy. The same group also demonstrated that exosomes containing TGF-β1 mRNA released by HG-treated GECs could activate mesangial cells in the glomerulus to promote renal fibrosis through the activation of TGF-β1/Smad3 signaling pathway [83]. 

### 4.2. EVs as Mediators of Fibroblast Activation in Renal Fibrosis

Various studies have pointed to EVs as important players in the process of fibroblast activation. Borges et al. demonstrated that hypoxic stress led to the secretion of exosomes from injured tubular epithelial cells, which subsequently activated kidney fibroblasts promoting their proliferation and extracellular matrix production [4]. The authors pointed to exosome-derived TGF-β1 mRNA as responsible for fibroblast activation. In another recent paper, the incubation of renal fibroblasts (NRK-49F) with exosomes from HG-treated PTECs induced proliferation and activation of fibroblasts with subsequent production of fibronectin, αSMA and collagen type I. Cell-to-cell communication mediated by PTEC-derived exosomes contributed to the onset of interstitial fibrosis in diabetic kidney disease (DKD) in which exosomal enolase 1 (Eno1) could play a pivotal role [84]. EVs from aldosterone-treated PTECs promoted fibroblast proliferation and migration, as well as increased expression of ECM proteins [85]. Furthermore, C57BL/6 mice administered with EVs derived from the renal cortex of aldosterone-treated db/db mice showed increased ECM production [85]. The authors proposed miR-196b-5p from EVs as responsible for EVs-mediated fibroblast activation and promotion of renal fibrosis in diabetic mice. Another evidence of epithelium–fibroblast communication in the pathogenesis of renal fibrosis was reported by Guan et al. [86]. The authors demonstrated that exosomes derived from hypoxia-injured PTECs could promote the activation and proliferation of fibroblasts in vitro. Injured PTECs successfully shuttled exosomes containing miR-150 to fibroblast cells in culture. Furthermore, mice affected by ischemic injury developed more pronounced renal fibrosis when administered with exosomes enriched with miR-150, making this exosomal miRNA an important mediator of fibroblast activation during renal fibrogenesis. Lu et al. [87] confirmed previously reported results on the role of PTEC-derived exosomes in the process of fibroblast activation, emphasizing the role of exosomal Shh ligand in the promotion of tubulointerstitial fibrosis.

### 4.3. EV-Mediated Recruitment of Inflammatory Cells in Renal Fibrosis

Recruitment and activation of inflammatory cells are of paramount importance for the processes of renal inflammation and fibrosis. Various stimuli can trigger activation of inflammatory cells during the process of kidney injury facilitating their infiltration and perpetuating the vicious cycle of tissue damage. Recent studies have shown that various neighboring or distant cells can communicate with inflammatory cells via extracellular vesicle release. Lv et al. [88] showed that TECs treated with albumin released exosomes that promoted tubulointerstitial inflammation and fibrosis by transferring detrimental signals from TECs to interstitial macrophages. The authors demonstrated that in the setting of proteinuric kidney disease, albumin triggered TECs to release exosomes carrying CCL2 mRNA, which was delivered to macrophages initiating their activation and migration [88]. Furthermore, exosomes released from HIF-1α-activated TECs triggered the pro-inflammatory phenotype of macrophages by transferring miR-23a and suppressing the activity of the ubiquitin editor A20 in vitro [89]. In vivo, exosomes derived from HIF-1α-activated TECs after being injected into injured renal parenchyma of mice led to an increased tubulointerstitial inflammation critical for the evolution of interstitial fibrosis [89].

It has been reported that during the development of DKD, the predominant macrophage phenotype is M1. M1 macrophages secrete inflammatory factors with a cytotoxic effect to glomerular mesangial cells and epithelial cells, thereby promoting the process of renal fibrosis in DKD [90]. Jia et al. [90] demonstrated that EVs secreted by renal proximal tubular epithelial (HK-2) cells after treatment with human serum albumin (HAS) induced macrophage M1 polarization in the presence of LPS. The authors found miR-199a-5p to be increased in EVs from HAS-induced HK2 cells and in urinary EVs from patients with diabetes mellitus (DM). Tail vein injection of DM mice with EVs from HAS-induced HK2 cells induced kidney macrophage M1 polarization and accelerated progression of DKD through miR-199a-5p and subsequently by targeting the Klotho/TLR4 pathway [90].

## 5. Extracellular Vesicles as Therapeutic Agents in Renal Fibrosis

An increasing body of evidence demonstrates a renoprotective effect of extracellular vesicles, especially those derived from mesenchymal stem (stromal) cells (MSCs). MSCs are multipotent stem cells, which can be isolated from different sources including bone marrow, umbilical cord, fat (adipose) tissue, or placenta (amniotic fluid) [91]. MSCs can self-renew by dividing and have the potential to differentiate into multiple cell types, including osteoblasts, chondrocytes, adipocytes, endothelial, cardiovascular, and neurogenic cell types [91]. It has been shown that MSCs have valuable anti-inflammatory and antifibrotic characteristics and an important role in tissue regeneration. Importantly, extracellular vesicles (EVs) derived from MSCs also have antifibrotic properties and have emerged as a powerful cell-free therapy for a variety of fibrotic states [7]. The administration of EVs derived from MSCs and other sources was found to be beneficial in different models of renal fibrosis (Table 1). 

### 5.1. Bone Marrow MSC-Derived EVs as Therapeutic Agents in Renal Fibrosis 

Several lines of evidence described the role for bone marrow mesenchymal stem cell (BM-MSC)-derived EVs as therapeutic agents in renal fibrosis. Namely, Wang et al. [92] demonstrated that human BM-MSC-derived exosomes carrying exogenous miR-let7c shuttled their cargo to TECs and attenuated renal fibrosis via targeting TGF-βR1. Zhou et al. [98] proposed a very interesting strategy to enhance the therapeutic capacity of BM-MSC-derived EVs. The authors demonstrated that a self-assembling peptide (KMP2) nanofiber hydrogel enhanced therapeutic potency of mouse BM-MSC-derived EVs in decreasing macrophage infiltration and chronic renal fibrosis in ischemia-reperfusion (I/R) injury in mice. The effect of EVs derived from human BM-MSC was also evaluated in a model of aristolochic acid-induced nephropathy (AAN) in mice. Of interest, intravenous administration of BM-MSC-EVs significantly ameliorated renal function, tubular kidney injury, and interstitial fibrosis in the AAN model. The authors proposed regenerative and antifibrotic roles for human BM-MSC-derived EVs in AAN through a transfer of different cargo molecules such as proteins or miRNA [93]. Moreover, EVs obtained from BM-MSCs reversed TGF-β1-induced morphological changes in HK-2 cells and restored kidney function in unilateral ureteral obstruction (UUO) in mice by miRNA-dependent repair mechanisms [94]. In addition, EVs derived from BM-MSCs [97,112] and bone marrow cells themselves [95] showed potential therapeutic effects against renal fibrosis through shuttling different miRNA cargo. In this context, Zhou et al. [95] proved that BM-MSC-derived EVs containing miR-144 when taken up by renal fibroblasts successfully inhibited activation of the tPA/MMP9-mediated proteolytic network and mitigated renal fibrosis. EVs shed by human BM-MSCs and by human liver stem-like cells (HL-MSCs) showed a therapeutic effect on progression and reversal of fibrosis in a mouse model of diabetic nephropathy (DN) induced by streptozotocin, while specific EV’s miRNA cargo was responsible for the modulation of fibrosis-related genes [96]. A protective role of HL-MSC-derived EVs has also been confirmed in an AA-induced mouse model of CKD, where treatment with HL-MSC-derived EVs reduced tubular necrosis, interstitial fibrosis, and infiltration of CD45 cells and fibroblasts [109].

### 5.2. Umbilical Cord MSC-Derived EVs as Therapeutic Agents in Renal Fibrosis 

Application of umbilical cord MSC-derived EVs has also been shown as a successful strategy to combat renal fibrosis in various models of disease. For instance, exosomes from human umbilical cord MSCs (UC-MSC) have been proposed as novel nanomaterials for regenerative medicine [99]. Ji et al. [99] demonstrated that infusion of human UC-MSC-derived exosomes promoted shuttling of nuclear YAP to the cytoplasm and attenuated collagen deposition and interstitial fibrosis in the kidney through the delivery of CK1δ and β-TRCP into recipient cells. Li et al. (2020) proposed the possible use of UC-MSC-shed exosomes as a novel cell-free therapeutic approach for renal fibrosis in DN. Namely, using GW4869, an inhibitor of exosome secretion, the authors demonstrated that UC-MSC-shed exosomes were responsible for the inhibition of myofibroblast transdifferentiation triggered by the TGF-β1/Smad2/3 signaling pathway and mesangial cell proliferation mediated by PI3K/Akt and MAPK signaling pathways in vitro [113]. The effect of UC-MSC-derived EVs was also evaluated in the I/R model of renal injury. Zhang et al. [100] demonstrated that human UC-MSC-derived EVs improved kidney function and ameliorated renal fibrosis progression, as well as Snail upregulation, in kidneys affected by I/R injury. Furthermore, the authors showed that overexpression of Oct-4 enhanced the therapeutic effect of EVs [100]. In addition, EVs derived from UC-MSCs [101,102] could elicit their potential therapeutic effect against renal fibrosis after I/R injury through various mechanisms. Namely, Zou et al. [102] demonstrated that UC-MSC-derived EVs had the potential to modulate the expression of renal VEGF and HIF-1α. In vitro, EVs delivered VEGF directly to TECs, thereby increasing VEGF levels. Consistently, a single administration of UC-MSC-derived EVs ameliorated renal fibrosis after I/R through suppression of C-X3-C motif ligand-1 (CX3CL1) [101].

### 5.3. Adipose Tissue MSC-Derived EVs as Therapeutic Agents in Renal Fibrosis 

Beneficial effects of extracellular vesicles derived from adipose tissue mesenchymal stem cells (AD-MSCs) on renal fibrosis have also been demonstrated in different models of disease. In a porcine model of metabolic syndrome and renal artery stenosis, a single intrarenal delivery of adipose tissue mesenchymal stem cell (AD-MSC)-derived EVs attenuated inflammation, oxidative stress, and renal fibrosis [103,104]. The renoprotective effect was elicited through shuttling of IL10 to recipient cells, while vesicles with pre-silenced IL10 were ineffective. Interestingly, labeled EVs were detected in the stenotic kidney (9% of the injected amount) on day 2 after injection and decreased thereafter, remaining at 2% by four weeks after injection [103]. The same group reported a role for AD-MSC-derived EVs in preserving kidney cellular integrity after renal injury through their effect on necroptosis [114]. Consistently, exosomes derived from human AD-MSCs mitigated renal fibrosis in mice after unilateral renal I/R injury via activation of tubular Sox9 [105]. In line with this observation, AD-MSCs-derived exosomes significantly ameliorated renal fibrosis score in mice subjected to UUO through activation of the SIRT1/eNOS signaling pathway [106]. 

### 5.4. EVs Derived from Other Sources as Therapeutic Agents in Renal Fibrosis 

Several lines of evidence point to the role of EVs derived from a healthy renal tubular cell in the protection against renal fibrosis. Namely, Dominguez et al. [107] showed that primary renal tubular cell-derived EVs alleviated renal fibrosis after renal I/R injury in rats and corrected renal transcriptome drift caused by kidney injury, as measured by RNAseq analysis. Moreover, intra-arterial injection of EVs derived from renal scattered tubular cells attenuated kidney fibrosis and improved renal function after unilateral renal arterial stenosis in mice [108]. Interestingly, EVs released from TECs treated with exendin-4 protected renal cells from fibrosis by decreasing miR-192 and inhibiting GLP1R downregulation in a p53-dependent manner in vitro [79]. Wang et al. [110] engineered an exosome vector containing Lamp2b fused with the peptide RVG, which targeted exosomes to organs that express the acetylcholine receptor, such as the kidney. Intramuscular injection of this kind of exosomes derived from muscle stem cells and containing miR-29 partially alleviated UUO-induced kidney fibrosis by targeting YY1 and TGF-β3. Of interest, EVs released from kidney-derived MSCs pre-incubated with RNase failed to ameliorate TGF-β1-induced peritubular capillary rarefaction and tubulointerstitial fibrosis in mice after UUO [111], confirming the potential therapeutic effect of EVs’ miRNA cargo.

## 6. EVs as a New Therapeutic Approach—Advantages and Perspectives 

Given all the presented evidence regarding functional roles of EVs and their beneficial effects in various models of renal fibrosis, it is conceivable that a new therapeutic approach for treatment/prevention of renal fibrosis, based on the use of EVs as target and/or tools, could be designed. 

EVs are generally considered to have great therapeutic potential due to their unique set of properties. Namely, EVs are natural, biocompatible products; they carry multiple information and are able to cross many biological barriers. They can be targeted and modified to carry drugs and/or target molecules. Importantly, EVs can replace cell therapy, i.e., MSCs, since they can convey cell benefits while having a critical advantage over the use of live cells due to their inability to proliferate and cause tumors [29,115,116]. However, on the odyssey toward a new therapeutic approach, many challenges still need to be overcome. 

One approach in this new line of therapy would be designing a therapeutic that would interrupt EV-based communication between cells in the processes leading to renal fibrosis by blocking the production or uptake of EVs. Studies of the release and uptake of EVs have identified several molecules involved in these processes and several compounds that could inhibit them. EV release can be inhibited along their different pathways of biogenesis by affecting EV trafficking or lipid metabolism as comprehensively reviewed by Catalano and O’Driscoll [117]. Thus, EV trafficking could be inhibited by calpeptin, manumycin A, or Y27632 [117]. Namely, calpeptin inhibits calpain, a family of calcium-dependent cytosolic cysteine proteases involved in cytoskeleton remodeling and consequently MV shedding, while manumycin A inhibits Ras farnesyltransferases, enzymes necessary for enabling the function of Ras. Y27632 competitively inhibits Rho-associated protein kinases (ROCK), which are enzymes involved in cytoskeleton rearrangements during MV production. Furthermore, lipid metabolism can be interfered by using pantethine, imipramine, or GW4869 [117]. Pantethine, a derivate of vitamin B5, inhibits cholesterol and total fatty acids synthesis and blocks translocation of phosphatidylserine to the outer membrane leaflet, thus inhibiting MV formation. Imipramine inhibits acid sphingomyelinase, an enzyme that catalyzes the hydrolysis of sphingomyelin to ceramide and has been shown to be important in both exosome and MVs biogenesis. GW4869 inhibits neutral sphingomyelinase, an enzyme that also generates ceramide. As for uptake inhibitors, they have also been reviewed elsewhere [29,33]. As reported by Mulcahy et al. [33], apart from blocking specific proteins involved in the recognition and interaction of EVs and target cell membrane by specific antibodies, there are numerous chemical compounds that inhibit uptake processes. Thus, cytochalasin D depolymerizes the actin, resulting in endocytosis inhibition; chlorpromazine inhibits clathrin-mediated endocytosis; filipin can deplete cholesterol necessary for caveolin-dependent endocytosis; dynasore blocks Dynamin2, which plays a role in membrane binding and membrane curvature; NSC23766 inhibits Rac1, a GTPase that has a major role in macropinocytosis; wortmannin inhibits PI3Ks, an important player in phagocytic processes, while simvastatin disrupts lipid raft-mediated endocytosis [33]. However, in an attempt to translate any of these EV release/uptake inhibitors into potential therapeutic agents, two main issues have to be considered. Firstly, the processes disrupted by these agents are important for cell functioning in general. Therefore, there are questions about dosage and toxicity. Additionally, it is worth noting that each compound may inhibit just one pathway and thus only affects the trafficking of a subset of EVs, which may diminish their efficacy. Secondly, even though they might not completely block EV release/uptake, we still do not know how important each EV population is for a particular process in different organs/physiological processes. Thus, even partial blocking of EV release/uptake on a systemic level for a prolonged period of time may be detrimental. Some of these compounds (e.g., calpeptin, pantethine, or GW4869) have been investigated in in vivo studies [118,119,120,121]. However, in most of these studies, inhibitors were not applied for a prolonged period of time, and no investigation of their systemic effects (apart from ones on target tissues) has been conducted. Therefore, we, as well as others [29,33,117], consider that the investigation of systemic effects of the identified inhibiting agents is a necessary step in the development of any therapeutic approach based on alterations of EV trafficking. Since most of the chemical compounds have no tissue specificity or are involved in a particular pathology, in this case, kidney disease, the issue of their targeting is, in our opinion, a critical one (Figure 3A). We believe that specific modifications of inhibiting agents designed to target EVs to specific cells involved in the development of renal fibrosis might be the way for the realization of this form of a novel therapy for renal fibrosis.

Alternatively, EVs themselves could be applied as therapeutic agents (Figure 3B). As stated above, MSC-derived EVs were demonstrated to be good candidates [29,103,122]. Although this kind of therapy has shown to be promising for many conditions, there are still a number of steps to be undertaken to make the transition from experimental to clinical application possible [123]. Issues such as large-scale production, quality control, stability, and standardization are common to all EV-based therapies and require the development of new/improved technologies [124,125]. Every specific type of therapeutic extracellular vesicle carries inevitable questions such as immunogenicity, toxicity, clearance, and influence on other cells and tissues. In relation to the latter, targeting and identification of active molecules are especially important issues [126]. The ultimate goal, as defined by theranostics, is to design a therapy, in this case, an EV-based one, which would have the capacity to recognize diseased cells and exert therapeutic action selectively on them [5]. EVs, as complex delivery and sensory yet natural system, provide the unique opportunity to be used for this purpose. 

Another approach of EV-based therapy would be a modification of EVs aimed at increasing their efficacy (Figure 3B). Thus, EVs can be loaded with drugs (synthetic or biological) or their originating cells can be engineered to produce EVs with desired properties (to carry specific surface or cargo molecules) [124,127]. One example of such an EV-based therapy could be the use of a vesicle that is able to target M1/M2 macrophage polarization in the kidney, thus reducing renal inflammation and facilitating kidney repair. For instance, exosomal miRNAs from MSCs, such as let-7b [128] and miR-182 [129], have been shown to be able to induce conversion of M1 macrophages to M2 phenotype and enhance tissue repair in different models of tissue injury in rodents [128,129]. One of the issues related to this approach is the efficacy of EV loading using current methods, which needs improvement [130]. 

The final approach encompasses a design of artificial EV-like vesicles, i.e., constructs consisting of a lipid-limiting membrane with targeting molecules on the surface of the vesicle and an active therapeutic cargo (Figure 3C) [124,126]. This approach would require the identification of both targeting molecules and active components. Having in mind the complexity of EV composition and signaling, singling out one active molecule might not be enough to reproduce the efficacy levels of natural EVs. Additionally, these constructs might be cytotoxic and/or immunogenic. However, due to the requirements of large-scale production and standardization, this line of investigation is very important. 

One of the often-mentioned issues regarding the use of EVs as therapy for renal diseases refers to the inability of EVs to pass the slit diaphragm due to a bigger size of the EV than the slit pore size [131,132]. Thus, it is not expected that any EV-based therapy would influence renal cells through glomerular blood vessels, except in pathologies that damage the glomerular filtration membrane. However, there are promising findings stating that intravenously administered EVs do reach renal cells and exert beneficial effects [106]. Notably, a quite small percentage of administered EVs reach the kidney, and this finding emphasizes the importance of resolving targeting issues when designing EV-based therapies. 

## 7. Conclusions

Extracellular vesicles are essential emissaries of information between cells and important players in various physiological and pathophysiological processes. EVs can enter blood vessels and reach different parts of the body transferring bioactive molecules such as proteins, RNAs, and lipids to distant target cells. It has been demonstrated that EVs play essential roles in tubulointerstitial communication during renal fibrosis since injured kidney cells may transfer EVs carrying various molecules to recipient cells, hence actively participating in renal inflammation and fibrosis.

One possible target of therapeutic EVs in renal fibrosis could be the renal tubular epithelial cell (TEC). After an injury, TECs display dramatic cellular and molecular drift that consequently leads to alterations of their epithelial secretome profile, making TECs important players in renal fibrosis. Injured TECs release EVs that can promote neighboring TECs to undergo EMT or activate fibroblasts and induce macrophage M1 polarization and their recruitment. EVs released from various MSCs and other cell sources have emerged as a powerful cell-free therapy in different models of renal fibrosis. MSC-derived EVs may suppress EMT of TECs, fibroblast activation, and myofibroblast transdifferentiation in vitro, shuttling different miRNAs and proteins. Furthermore, administration of MSC-derived EVs ameliorates tubular injury and interstitial fibrosis in different animal models of renal fibrosis.

Targeting EVs directly to inhibit their pathogenic effect or take advantage of their potential regenerative roles could be some of the promising therapeutic strategies. On an odyssey toward new therapeutic approaches for renal fibrosis lay numerous issues to be resolved; nevertheless, both the nature of EVs and the widespread efforts to make EV-based therapies in different areas, render EVs a promising tool in designing a new generation of therapy for renal fibrosis.

## Figures and Tables

**Figure 1 ijms-22-03887-f001:**
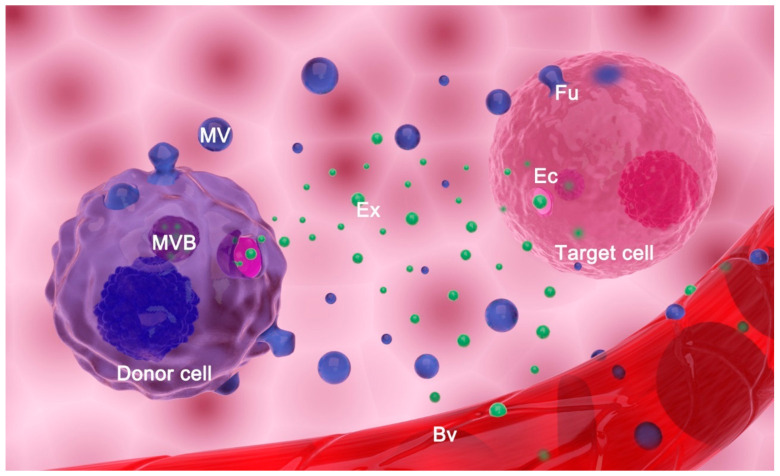
Extracellular vesicles (EVs) release and transfer. EVs comprise exosomes (Exs), microvesicles (MVs), and apoptotic bodies (not shown). Exs are released from a donor cell upon fusion of a multivesicular body (MVB) with the plasma membrane. Microvesicles (MVs) are formed by budding of the originating cell’s plasma membrane. EVs reach target cells of the same tissue or they are transferred by body fluids to distant cells (i.e., EVs enter blood vessels (Bv) and are transferred by blood to distant organs/tissues/cells). EVs may deliver information to the target cell by (a) fusion (Fu) with its plasma membrane; (b) endocytosis (Ec) of EVs by target cells; or (c) interaction of surface molecules on EVs and the cell (not shown).

**Figure 2 ijms-22-03887-f002:**
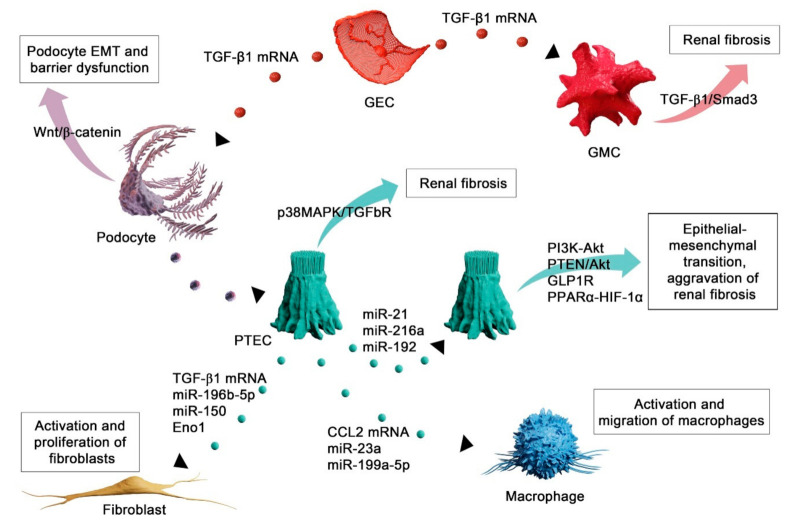
EV-mediated crosstalk between various renal cellular compartments during renal fibrosis. Injured renal proximal tubular epithelial cells (PTECs) can release and transfer EVs carrying different cargo to neighboring PTECs promoting epithelial–mesenchymal transition (EMT) and progression of renal fibrosis via different signaling pathways. Of note, injured podocytes can also transfer EV cargo to PTECs, inducing a strong pro-fibrotic response in these cells. Furthermore, EVs derived from injured PTECs participate in the process of fibroblast and macrophage cell activation, shuttling different cargo molecules into the recipient cell. Injured GECs shuttle their EVs containing TGF-β1 mRNA to podocytes, mediating EMT and barrier dysfunction, as well as activation of GMCs. PTEC, proximal tubular epithelial cell; GEC, glomerular endothelial cell; GMC, glomerular mesangial cell.

**Figure 3 ijms-22-03887-f003:**
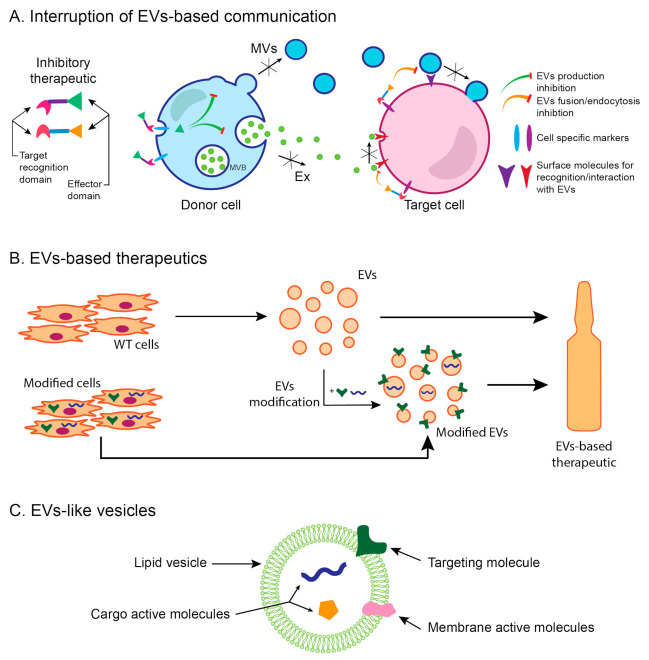
EV-targeted/based therapeutic approaches. (**A**) In the first approach, an interruption of EV-based communication between cells can be achieved by different chemical compounds that inhibit specific pathways of EV biogenesis or uptake. However, those compounds may exert their effects in a nonselective manner, thus affecting every cell in the organism that comes in contact with them. This issue could be resolved by combining chemical compounds with targeting moieties. (**B**) In the second approach, EVs can be used as therapeutics, either in their native form (such as MSC-EVs) or after their modification (loading with drugs, miRNA, and/or target molecules). These modifications can be achieved by direct manipulation of isolated EVs or by engineering the cells that produce them. (**C**) The third approach represents a design of EV-like vesicles consisting of lipid bilayer vesicles carrying targeting molecules and active components as cargo or membrane molecules.

**Table 1 ijms-22-03887-t001:** Application of EVs as therapeutic agents in renal fibrosis.

EV Source	In Vivo Model of Renal Fibrosis	EV Cargo	Signaling Pathway	EVs Administration	Reference
BM-MSCs	UUO	miR-let7c	TGF-βR1	Intravenous	[92]
	AAN	miRNAs	LTBP1/TGF-β1	Intravenous	[93]
	UUO	miRNAs	n/i	Intravenous	[94]
	UUO	miR-144	tPA/MMP9	Intravenous	[95]
	STZ-DN	miRNAs	Snail, FAS	Intravenous	[96]
	UUO	miRNAs	n/i	Intravenous	[97]
	I/R injury	n/i	n/i	Intrarenal	[98]
UC-MSCs	UUO	CK1δ, β-TRCP	YAP	Intravenous	[99]
	I/R injury	Oct-4	Snail	Intravenous	[100]
	I/R injury	n/i	CX3CL1	Intravenous	[101]
	I/R injury	RNAs, VEGF	n/s	Intravenous	[102]
AD-MSCs	MS+AS	IL10	n/i	Intrarenal	[103,104]
	I/R injury	n/i	Sox9	Intravenous	[105]
	UUO	GDNF	SIRT1/eNOS	Intravenous	[106]
RTCs	I/R injury	RNAs	n/i	Intravenous	[107]
R-STCs	UUO	Mitochondrial	Mitochondrial	Intra-arterial	[108]
	STZ-DN	miRNAs	Snail, FAS	Intravenous	[96]
HL-MSCs	AAN	n/i	multiple	Intravenous	[109]
MusSCs	UUO	miR-29	YY1, TGF-β3	Intramuscular	[110]
K-MSCs	UUO	miRNAs	n/i	Intravenous	[111]

MSCs, mesenchymal stem cells; BM-MSCs, bone marrow MSCs; UC-MSCs, umbilical cord MSCs; AD-MSCs, adipose tissue MSCs; RTCs, renal tubular cells; R-STCs, renal scattered tubular cells; HL-MSCs, human liver stem-like cells; MusSCs, muscle stem cells; K-MSC, kidney-derived MSCs; UUO, unilateral ureteral obstruction; I/R, ischemia-reperfusion; AAN, aristolochic acid-induced nephropathy; STZ-DN, streptozotocin-induced diabetic nephropathy; MS+AS, metabolic syndrome and renal artery stenosis; n/i, not investigated; n/s, not specified.

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
