# Peer review of "Extracellular Vesicles and Renal Fibrosis: An Odyssey toward a New Therapeutic Approach"

_ijms, 2021, doi:10.3390/ijms22083887_

Round 1
Reviewer 1 Report
In their review, Kosanović and other review current understanding of the role of EVs in renal fibrosis pathology as well as their utility as therapeutic agents. Overall, the review is elegantly and clearly written, and the ideas are very well articulated. The review is impressively comprehensive and the covers the latest work done in the field of renal fibrosis and emerging understanding of the role of EVs in it. A few minor comments for the authors to consider.
- The two main classes of EVs are generally exosomes and ectosomes (under which both microvesicles and apoptosomes are classified).
- In section 6, a more in depth discussion of EV inhibition is desired. There are many small molecule candidates being tested for EV inhibition including GW4969, Rab-inhibitors and other blockers of the ESCRT pathway. There are also gene silencing strategies. Authors should mention actual studies that support their claim that this strategy would otherwise be detrimental to kidney function (as intuitive as such a claim might be).
- The review should also cover the utility of EVs as diagnostic and prognostic tools.
- It would be helpful to summarize the therapeutic (and diagnostic) roles of EVs in a figure.
Author Response
Reviewer 1
In their review, Kosanović and other review current understanding of the role of EVs in renal fibrosis pathology as well as their utility as therapeutic agents. Overall, the review is elegantly and clearly written, and the ideas are very well articulated. The review is impressively comprehensive and the covers the latest work done in the field of renal fibrosis and emerging understanding of the role of EVs in it. A few minor comments for the authors to consider.
We thank the Reviewer #1 for valuable comments and suggestions that we think significantly improved the quality of the manuscript.
Reviewer 1, comment 1: The two main classes of EVs are generally exosomes and ectosomes (under which both microvesicles and apoptosomes are classified).
Response: We thank the Reviewer #1 for pointing out the issue of nomenclature. Regarding the terminology in our manuscript, we have followed ISEV guidelines (https://www.tandfonline.com/doi/full/10.1080/20013078.2018.1535750), except in the case of citing original authors’ statements from their publications. To our knowledge, and carefully reading the MISEV guidelines and other relevant papers such as “Ectosomes” Emanuele Cocucci and Jacopo Meldolesi, Current Biology Vol 21 No 23R940 (https://doi.org/10.1016/j.cub.2011.10.011), we cannot see that apoptotic bodies are commonly classified as ectosomes. Therefore, we would prefer not to introduce this classification in the manuscript.
Reviewer 1, comment 2: In section 6, a more in depth discussion of EV inhibition is desired. There are many small molecule candidates being tested for EV inhibition including GW4969, Rab-inhibitors and other blockers of the ESCRT pathway. There are also gene silencing strategies. Authors should mention actual studies that support their claim that this strategy would otherwise be detrimental to kidney function (as intuitive as such a claim might be).
Response: We thank Reviewer #1 for this useful comment and we agree that a more detailed discussion of EV release inhibition would be helpful for better understanding its potential therapeutic effect. Thus, we have introduced additional information in the Section 6, pages 11-12. Also, we have now clarified that we think systemic effects (not just the ones detrimental to kidney function) of such therapies need assessment as a necessary step in EV-inhibition translational investigation. (Section 6, page 12).
Reviewer 1, comment 3: The review should also cover the utility of EVs as diagnostic and prognostic tools.
Response: The main purpose of our review paper was to provide an overview of the renoprotective role of EVs from diverse sources and their use as therapeutic agents in renal fibrosis, highlighting benefits and new perspectives of therapeutic applications of EVs in renal diseases. The utility of EVs as diagnostic and prognostic tools in different kidney diseases fall outside the scope of this paper and has already been extensively review elsewhere (PMIDs: 25688242, 27376269; 29081510; 27582107; 26251351). Nevertheless, we have now included additional information explaining that EVs can be used as diagnostic and prognostic tools in different renal diseases (Section 2.2, page 4).
Reviewer 1, comment 4: It would be helpful to summarize the therapeutic (and diagnostic) roles of EVs in a figure.
Response: We thank the Reviewer #1 for this valuable suggestion. We have now included the schematic summary of different therapeutic approaches mentioned in the Section 6, presented as Figure 3 (Section 6, page 13).

Reviewer 2 Report
The authors provide a thorough review on the potential role and use of EV in renal fibrosis. They provide the most recent research on this field and an overview about their potential use as therapeutic agents. Therefore I consider that this work may be published in its present form.
Author Response
Reviewer 2
The authors provide a thorough review on the potential role and use of EV in renal fibrosis. They provide the most recent research on this field and an overview about their potential use as therapeutic agents. Therefore I consider that this work may be published in its present form.
Response: We thank the Reviewer #2 for the positive appraisal of our work.

Reviewer 3 Report
In this review Kosanović and colleagues provide an overview of the role played by extracellular vesicles in kidney injury and, on the other hand, a possible renoprotective effect of specific kinds of vesicles.
The article is up-to-date and reflects what can be found in other reviews on the argument.
I can give a couple of small suggestions:
The authors may consider adding a few more lines about diagnostic markers in urinary exosomes (that are already referenced) such as miRNA-145 and -192, protein markers etc.
Since the detrimental M1 macrophage polarization is mentioned the authors may look for EV-based strategies for M2 switching. Again, some microRNAs are often reported in literature.
Author Response
Reviewer 3
In this review Kosanović and colleagues provide an overview of the role played by extracellular vesicles in kidney injury and, on the other hand, a possible renoprotective effect of specific kinds of vesicles.
The article is up-to-date and reflects what can be found in other reviews on the argument.
I can give a couple of small suggestions:
Response: We thank the Reviewer #3 for the positive assessment of our work.
Reviewer 3, comment 1: The authors may consider adding a few more lines about diagnostic markers in urinary exosomes (that are already referenced) such as miRNA-145 and -192, protein markers etc.
Response: We have now included additional information explaining that EVs can be used as diagnostic and prognostic tools in different renal diseases and mentioned different miRNAs (such as miR-145 and -192), mRNA and proteins from urinary EVs as potential diagnostic markers, as requested (Section 2.2, page 4). We did not focus extensively on EVs as diagnostic and prognostic tools as this subject was out of the scope of this paper and has already been reviewed in details elsewhere (PMIDs: 25688242, 27376269; 29081510; 27582107; 26251351).
Reviewer 3, comment 2: Since the detrimental M1 macrophage polarization is mentioned the authors may look for EV-based strategies for M2 switching. Again, some microRNAs are often reported in literature.
Response: We thank the Reviewer #3 for this suggestion. We have now added additional information on EV-based strategy for M2 switching in the Section 6, page 12.
